# Effects of Chest Physiotherapy in Preterm Infants with Respiratory Distress Syndrome: A Systematic Review

**DOI:** 10.3390/healthcare11081091

**Published:** 2023-04-11

**Authors:** Ana Igual Blasco, Jessica Piñero Peñalver, Francisco Javier Fernández-Rego, Galaad Torró-Ferrero, Julio Pérez-López

**Affiliations:** 1International School of Doctorate of the University of Murcia (EIDUM), University of Murcia, 30100 Murcia, Spain; 2Early Intervention Center Fundación Salud Infantil, 03201 Elche, Spain; 3Nebrija Center for Research in Cognition of Nebrija University (CINC), Nebrija University, 28015 Madrid, Spain; 4Department of Developmental and Educational Psychology, Faculty of Psycology, University of Murcia, 30100 Murcia, Spain; 5Research Group in Early Intervention of the University of Murcia (GIAT), University of Murcia, 30100 Murcia, Spain; 6Department of Physical Therapy, Faculty of Medicine, University of Murcia, 30100 Murcia, Spain

**Keywords:** respiratory distress syndrome, physical therapy modalities, premature and neonatal intensive care unit

## Abstract

Preterm birth carries a higher risk of respiratory problems. The objectives of the study are to summarize the evidence on the effect of chest physiotherapy in the treatment of respiratory difficulties in preterm infants, and to determine the most appropriate technique and whether they are safe. Searches were made in PubMed, WOS, Scopus, Cochrane Library, SciELO, LILACS, MEDLINE, ProQuest, PsycArticle and VHL until 30 April 2022. Eligibility criteria were study type, full text, language, and treatment type. No publication date restrictions were applied. The MINCIR Therapy and PEDro scales were used to measure the methodological quality, and the Cochrane risk of bias and Newcastle Ottawa quality assessment Scale to measure the risk of bias. We analysed 10 studies with 522 participants. The most common interventions were conventional chest physiotherapy and stimulation of the chest zone according to Vojta. Lung compression and increased expiratory flow were also used. Heterogeneities were observed regarding the duration of the interventions and the number of participants. The methodological quality of some articles was not adequate. All techniques were shown to be safe. Benefits were described after conventional chest physiotherapy, Vojta’s reflex rolling, and lung compression interventions. Improvements after Vojta’s reflex rolling are highlighted in the comparative studies.

## 1. Introduction

Neonatal Respiratory Distress Syndrome (RDS) is a condition of pulmonary insufficiency that begins at or shortly after birth and increases in severity during the first 2 days of life. Clinically, RDS is present with early respiratory distress including cyanosis, grunting, retractions, and tachypnoea. If therapeutic measures are not applied, the child could potentially die from progressive hypoxia and respiratory failure. RDS is due to a deficit of alveolar surfactant together with a structural immaturity of the lungs [1]. To achieve effective gas exchange, it is necessary to have a large lung surface area and a thin alveolar-capillary barrier.

In preterm birth, babies present structural and functional immaturity. In addition, the exposure to routine interventions aimed to increase survival, such as oxygen supplementation and mechanical ventilation, can produce lung injuries and interrupt further lung development [2]. The risk of suffering RDS is inversely proportional to gestational age and birth weight [3]. Its incidence is around 60% in babies under 28 weeks of gestation (wg) and less than 5% in those over 34 wg [4]. The goal of RDS treatment is to provide interventions to maximise survival while minimising potential adverse effects, including Bronchopulmonary Dysplasia (BPD).

BPD continues to be one of the most important chronic morbidities affecting very preterm infants [5], being considered the main cause of long-term morbidity and mortality among premature infants with low birth weight. Its incidence has not decreased in recent decades, described in around 40% in children with extremely low birth weight [6] and 43% in premature children of less than 28 wg [7]. Among the effects of DBP in premature infants, changes in lung structure and function have been observed in childhood [8], adolescence [9] and adulthood [10]. The long-term consequences not only affect respiratory function, but also have an impact on neurological development [11] and are a risk factor for cerebral palsy [12].

Among the current proposed treatment methods for RDS and BPD, respiratory physiotherapy is required [13], which has techniques for unblocking both upper and lower airways [14], as well as techniques aimed to improve the ventilatory pattern. Taking into account the consequences that RDS can have, the derived problems and its high prevalence, the objective of this systematic review is to know the different physiotherapy modalities in the treatment of RDS in premature new-borns. The aim is also to know which is the most appropriate and safe modality for the treatment of RDS in premature births. On the other hand, we assessed the risk of bias and the methodological quality of the studies included in the review, to determine which physiotherapy procedures can be applied with greater guarantee in the treatment of RDS.

## 2. Materials and Methods

**Registration and protocol:** A review protocol was developed in accordance with the PRISMA (Preferred Reporting Items for Systematic Reviews and Meta-analysis) guidelines. The protocol can be found on PROSPERO. PROSPER ID: CRD42021246232. Registered on 1 April 2021.

### 2.1. Eligibility Criteria

We selected Randomized Clinical Trials (RCTs), controlled clinical trials, cohort or case-control studies, or quasi-experimental trials. No restrictions were made regarding the date; on whether we could access the full text in Spanish or English; or on those which interventions were chest physiotherapy interventions performed in intensive care units in preterm infants with RDS or respiratory distress associated with prematurity.

### 2.2. Information Sources

The databases consulted were: PubMed, Web of Science, Scopus, Cochrane Library, SciELO, LILACS, MEDLINE, ProQuest, PsycArticle and Virtual Health Library of Spain (VHL). Finally, different experts in the field were contacted to complete the identification of essential articles.

The search was carried out from April 2021 to April 2022. The last search in all databases was on 30 April 2022.

For data collection, the following search strategy was carried out: (“respiratory distress syndrome” OR “neonatal respiratory distress syndrome” OR “lung diseases” OR “bronchopulmonary dysplasia”) AND (“Chest Physiotherapy” OR “Respiratory physiotherapy” OR “cardiopulmonary physiotherapy” OR “physical therapy modalities”) AND (“Preterm infants” OR “premature”) AND (“neonatal” OR “neonates” OR “newborn”).

No limits were applied in the searches, except for the ProQuest database where we applied the scientific journal filter within the type of sources and for the type of article, we selected article, main article, and case study.

### 2.3. Study Selection Process

After the search and elimination of duplicate articles, two reviewers checked the title and abstract initially and the full text later, to eliminate those that did not meet the eligibility criteria. In case of disagreement, a meeting was held between both researchers to reach a consensus on their decision. If disagreement persisted, a third reviewer resolved potential discrepancies.

### 2.4. Data Extraction Process

One investigator from the team collected the data for each article, and another reviewer checked the data. Disagreements were resolved via consensus. In the event of disagreement, a third investigator was expected to make the decision.

Information was collected from each study on the type of study, the number of participants, the sample loss, the number of groups, the characteristics of the participants, the type of intervention, measurement instruments used, evaluation times, results and year of publication.

### 2.5. Assessment of the Risk of Bias and Methodological Quality of the Studies

The Methodology and Research in Surgery (MINCIR) scale [15] was used to measure the methodological quality of all articles, and the PEDro scale [16] for the RCTs. The Newcastle-Ottawa Scale (NOS) [17] was used to measure the risk of bias in observational studies and the Cochrane ROB-2 scale [18] in RCTs. Two independent investigators performed these evaluations. For the non-overlapping reference where the reviewers did not agree, a meeting was arranged to reach a common final decision. A third independent reviewer was in charge of making the final decision in case of persistent disagreement.

## 3. Results

### 3.1. Selection of Studies

After the bibliographic search in the different electronic databases and by consultation with experts, 267 initial studies were identified (22 in PubMed, 32 in Web of Science, 25 in Scopus, 8 in the Cochrane Library, 1 in SciELO, 2 in LILACS, 22 in MEDLINE, 136 in ProQuest, 0 in PsycArticle, 13 in VHL and 6 by consultation with experts). Of these, 84 articles were discarded due to duplicity, leaving 183 articles. After reading the title and abstract, 145 new articles were initially discarded for not meeting the eligibility criteria for article type, full text availability, and language. By contacting the authors, an attempt was made to recover the articles excluded because the full text was not available [19,20,21], but this action was unsuccessful, leaving 38 articles for full text revision. Subsequently, 28 articles were eliminated for not meeting the type of intervention criteria [22,23,24,25,26,27,28,29,30,31,32,33,34,35,36,37,38,39,40,41,42,43,44,45,46,47,48,49], leaving 10 articles to review (Figure 1).

### 3.2. Characteristics of the Studies

Ten articles [50,51,52,53,54,55,56,57,58,59] published in English that met the eligibility criteria were analysed. A total of 50% were RCTs [50,51,53,54,57], 40% were quasi-experimental before-after studies [52,55,58,59] and 10%, 1 article, was a retrospective and analytical case-control study [56]. The years of publication of the included studies ranged between 2006 and 2019.

### 3.3. Participants

The total number of patients analysed was 522. No distinction was made between the total number of individuals assigned to the control and experimental groups, since the distribution of the groups was different for each study and the same intervention modality was used as the main intervention in some studies and as a control in others. Metha, et al. [55] had two groups with the same intervention where the difference between the groups was a medical situation; each group had 30 participants. The study by Giannantonio, et al. [52] included two groups, one where the intervention was performed in the first week of life, 21 participants, and the other after the first week of life, 13 participants. Camy, et al. [50] also had two groups where the difference was the application of the intervention at two different times of the day. Both groups had 9 participants each. Wong, et al. [59] and Singh, et al. [58] only had one group where the intervention was applied, with a sample size of 11 and 42, respectively. Oliveira, et al. [56] had a control group without intervention of 61 participants and an experimental group of 93 participants. Kole, et al. [54] had three groups of 20 participants each, one control group and two experimental groups. Gharu, et al. [51], El-Shaarawy, et al. [57] and Kaundal, et al. [53] had a control group (30, 18 and 28 participants, respectively) and an experimental group (20, 18 and 28 participants, respectively).

It should be noted that the control group of the last 4 studies mentioned received an intervention modality that in the studies by Metha, et al. [55], Oliveira, et al. [56] and Singh, et al. [58] was considered as an experimental intervention modality.

All study participants are premature with less than 37 wg. All authors considered respiratory distress associated with prematurity as an inclusion criterion, through the diagnosis of RDS, hyaline membrane disease or the need for mechanical ventilation for more than 24 h without other serious adjacent health complications.

### 3.4. Characteristics of Physiotherapy Treatments

Table 1 describes the characteristics of the physical therapy of the 10 studies included in the intervention. In five studies [51,52,53,54,57], they used the first phase of reflex rolling according to Vojta (RRo) (one of them [51] used the first phase of RRo plus reflex creeping (RC)), described by Vojta in 1959 [60]. One study used Expiratory Flow Increase (EFI) [50], defined by Barthe in 1973 [61]. Two used pulmonary compression (PC) [54,59], defined by Wong in 1998 [62]; of these two studies, one [54] has two experimental groups (one group with Vojta’s reflex rolling and the other lung compression). In addition, three studies [55,56,57] used conventional chest physiotherapy (CPT) as the main intervention. In these studies that used CPT, the techniques were not homogeneous, although they all applied vibration and percussion techniques.

In the case of the control groups, a greater variety was observed. In the study by Oliveira, et al. [56], the control group received conventional neonatal care, and Gharu, et al. [51], Kaundal, et al. [53], Kole, et al. [54] and El-Shaarawy, et al. [57] used CPT for their control group.

The duration of the intervention was between 1 and 14 days, except for Oliveira, et al. [56], who does not mention the exact days. It mentions that the intervention was carried out while the babies were hospitalized and the intervention was carried out according to the criteria of the physical therapist. El-Shaarawy, et al. [57], on the other hand, mentions that the intervention was performed until the patient was discharged from the neonatal intensive care unit (NICU). The frequency of intervention was between one and three times per day, except in the article by Oliveira, et al. [56] who did not specify the frequency. The intensity of the intervention varied between 10 and 30 min, depending on the technique used. Finally, all studies were carried out by a physical therapist, within the NICU or neonatal care units of different hospitals.

### 3.5. Outcome Measures

All the studies carried out an assessment at the beginning and at the end of the intervention, with the exception of Oliveira, et al. [56], since it is a retrospective study and the measurements of the variables did not involve comparison with each other, but rather the duration in days of the recorded results. Several of the studies performed different consecutive assessments during the intervention or minutes afterwards, but no long-term follow-up assessments are observed in any of the cases. The study by Gharu, et al. [51] mentions long-term effects, but the comparison of the assessments in this study is between the first and last day of the intervention, as in other articles in this review.

The evaluations carried out in the different studies were mainly measurements related to respiratory function. As described in Table 2, very heterogeneous outcome measures were observed between studies. Six studies [51,52,53,54,55,57] recorded oxygen saturation (SpO_2_); four, respiratory rate (RR) [52,53,55,57]; three, duration of oxygen therapy [54,56,57]; two, respiratory system compliance [58,59]; and two, chest radiography [54,55] to evaluate the radiographic findings. The rest of the observed variables were measured in one article (Table 2).

### 3.6. Risk of Bias in Studies

The results of the methodological quality measurement through the MINCIR and PEDro scales are shown in Table 3 and Table 4, respectively. Moreover, the results of the risk of bias through the NOS and Rob-2 scale are shown in Table 5 and Table 6, respectively.

The MINCIR scale (Table 3) was applied to the 10 articles. The lowest total score was 15, which was shown in three articles [50,52,58]. Six articles presented a score of 16 [51,53,54,55,57,59], and one article presented a score of 23 [56], this being the highest score. Therefore, 10% presented an adequate methodological quality and 90% an inadequate methodological quality according to the MINCIR scale.

The PEDro scale (Table 4) was applied to five items. Items 5 and 10 were met in all studies and item 6 in none. The lowest score observed was 5 [51], with two articles showing a score of 6 [50,51] and another two of 7 [54,57]. We concluded that 80% of these articles have a good methodological quality [50,51,54,57] and 20% have a fair methodological quality [51] according to the PEDro scale. No article presented poor methodological quality.

The NOS scale (Table 5) was applied to an article [56] obtaining a score of 7 stars, which coincides with low risk of bias.

Five articles [50,51,54,57] were analysed with the Cochrane ROB-2 scale (Table 6), 60% (3 articles) showed some concerns [50,53,54] and 40% (2 articles) a high risk of bias [51,57]. All studies showed a low risk of bias regarding outcome measurement (100%). A total of 80% of articles showed low risk of bias for deviations from intended interventions and missing outcome data. Some concerns are observed in 100% of the articles regarding the selection in the reported result.

### 3.7. Individual Study Results

All studies that measured SpO_2_ [51,52,53,54,55,57] showed a significant intragroup improvement in this parameter, regardless of the technique applied. Regarding the RR, Mehta, et al. [55] showed a significant improvement after the application of CPT. Giannantonio, et al. [52] did not show a significant difference in the values of RR when applying the RRo Vojta in the first or second week of life. El-Shaarawy, et al. [57] and Kaundal, et al. [53] observed a significant decrease in RR both in the group that applied CPT and in the one that applied CPT + RRo Vojta, although no significant improvement was observed between the groups. The studies that measured the duration of oxygen therapy were El-Shaarawy, et al. [57] and Kole, et al. [54] who showed a significant difference in the decrease in days of oxygen therapy application in the group that received CPT + RRo Vojta compared to the rest of the groups. In the case of Oliveira, et al. [56] (applying CPT), a significant difference was found in favour of the group that did not have physiotherapists in the NICU, but it should be noted that the groups presented a significant difference regarding the weeks of gestation at birth and SDR among others. Respiratory system compliance was measured by Wong, et al. [59], applying PC, and Sing, et al. [58], applying CPT; in both cases, a significant improvement of this outcome measure was observed after the intervention. Mehta et al. [55] and Kole et al. [54] performed chest radiography during the study. The first study, using CPT, showed a significant improvement in posting radiological findings and no rib fractures, and the second, using CPT, PC, and RRo Vojta depending on the group, showed re-expansion of the collapsed areas. Hospitalization time in the NICU was measured by El-Shaarawy, et al. [57] who observed a significant reduction in days in the group that had received RRo Vojta compared to the group that only received CPT. Finally, Oliveira, et al. [56], applying CPT, did not observe a significant difference in the days of hospitalization between their groups.

The rest of the recorded parameters were measured in only one of the studies (see Table 2).

In the case of Mehta, et al. [55], they also measured heart rate (HR); the Silverman Anderson scale score (SA), where it decreased significantly after 15 min (5 min after application); and auscultation, where there was a significant reduction in crepitus after suction. Giannantonio, et al. [52] also measured the transcutaneous oxygen pressure where a significant difference was observed between the values of the different measurements. He performed a brain ultrasound where it was concluded that in no case did periventricular leukomalacia appear or worsen the intraventricular haemorrhage. Premature Infant Pain Profile (PIPP) and Neonatal Infant Pain Scale (NIPS) showed no pain or stress. Camy, et al. [50] measured the gastroesophageal reflux index and concluded that it does not increase. Oliveira, et al. [56] measured the duration of need for CPAP, the duration of need for invasive mechanical ventilation, the duration of need for non-invasive mechanical ventilation and the duration of need for non-invasive mechanical ventilation plus CPAP, and observed a significant difference between longer duration of ventilatory support in the group that had a physiotherapist in the NICU. Wong, et al. [59] measured the resistance of the respiratory system where no significant changes were observed after the application of PC. Kole, et al. [54] assessed the partial pressure of oxygen, which improved significantly after the interventions in all groups, and the treatment time, where fewer days of intervention were recorded in the PC group. Singh, et al. [58] also measured the re-expansion of collapsed lobes and the reduction of inspiratory and expiratory resistance, where significant differences were observed in all outcome measures.

## 4. Discussion

### 4.1. Evidence Summary

Regarding the characteristics of the sample, it is worth noting the differences that exist on the need for ventilatory support at the time of the intervention. Although all the studies are included within the diagnosis of RDS, hyaline membrane disease or the need for initial mechanical ventilation, not all of them had the same inclusion criteria regarding ventilatory support. Mehta, et al. [55] initially differentiated the two groups between participants mechanically ventilated and those without this support. Camy et al. [50] reported that the participants with MV or oxygen therapy for more than 28 days had a diagnosis of BPD, but did not specify whether they continued with this support at the time of the intervention. Similarly, Oliveira et al. [56] described ventilatory support for more than 24 h as an inclusion criterion, but it is not described throughout the intervention. Kaundal, et al. [53] does not describe whether they are carriers of ventilatory support; they do highlight the diagnosis of BPD, but they did not specify such support at the time of the intervention. Finally, Gharu et al. [51] specified that they should be non-ventilated participants. On the contrary, the rest of the studies [52,54,57,58,59] described some type of ventilatory support. This is an aspect to take into account, since the initial medical situation is not homogeneous with respect to this parameter, similar to the heterogeneity of the results measured between the different studies, in the different intervention techniques applied and in the period of application, frequency and intensity. These aspects can hinder the homogeneity of results and, therefore, the drawing of conclusions.

Regarding intervention techniques, the objectives of CPT revolve around the elimination of excess secretions, treating pulmonary collapse, reducing reintubations, helping pulmonary ventilation, etc. In the NICU, this usually involves positioning, percussion, vibrations, saline instillation, oropharyngeal or endotracheal aspiration, and manual hyperinflation [63]. It is worth noting the benefits observed regarding the reduction of respiratory [53,55,57] and HR [55] and the increase in SpO_2_ [51,53,54,55,57], where significant differences are described. Moreover, Mehta, et al. [55] shows the decrease in crackles just after suction, although the latter is not maintained over time, an aspect that supports the need for 24-h intervention [55]. On the other hand, there is an improvement in the compliance of the respiratory system [58,59] and a reduction in inspiratory and expiratory resistance [58]. Similarly, Santos, et al. [64] in 2009, analysed 18 premature children divided into two groups according to the days of MV (+/− 5 days). All of them underwent a chest physiotherapy session. They obtained similar results that support these benefits.

Although all the studies analysed in this review, which applied CPT, used percussion and vibration during the intervention, the interventions were not homogeneous. Mehta, et al. [55] and Singh, et al. [58] accompanied these two techniques with suction. In addition, in the first study, the baby was placed in the prone position, and the second study describes that nebulization and hyperinflation with an Ambu bag in the lateral decubitus position. On the other hand, Oliveira, et al. [56] did not describe exactly which CPT techniques he applied to each participant in the experimental group. Finally, the application period was uneven, and in some of them, the technique was applied as the main intervention [55,56,58,59], and in others, as a control [51,53,54,57]. Although there is some evidence in favour of CPT, it is considered insufficient. Hough, et al. [65] concludes that there is insufficient evidence to base clinical practice using CPT, and RCTs with larger sample sizes are needed. Based on these arguments, in the present review, we find that CPT improves outcome measures in intragroup analyses, whether acting as the control or experimental group. If we analyse the cases in which the CPT acts as the control [51,53,54,57], in three of the four studies [51,53,57], despite being improvements in the different groups, this is greater in the experimental group in which Vojta therapy is applied.

The studies that applied Vojta therapy present a more homogeneous intervention because it is a defined technique [60]. In all cases, the first phase of Vojta’s RRo was applied, except for the study by Gharu, et al. [51] who also applied Vojta’s RC. The Vojta principle applies isometric strengthening techniques through tactile and proprioceptive stimulation, to activate the typical and ideal development of patterns of posture and movement, and therefore to improve breathing patterns [66]. Sanz-Esteban, et al. [67] in 2021 states, in a study carried out in healthy adults, that specific tactile and proprioceptive sensory stimulation in the intercostal space and in the mammillary line between ribs 7 and 8, according to Vojta therapy, evokes the activation of cortical areas with great influence on motor planning and execution. This activation of the central nervous system through Vojta therapy manifests innate patterns of posture and movement, including respiratory ones. These results obtained in adults using Vojta therapy could be extrapolated to the neonatal population. In this line, in the present review, benefits are reflected in SpO_2_ [51,52,53,54,57], transcutaneous oxygen pressure (PtcO_2_) [52], and partial pressure of oxygen (PaO_2_) and lung re-expansion [54]; no negative effects are observed in transcutaneous carbon dioxide pressure (PtcCO_2_) [52] and RR [52,57]. Linking these results could confirm that the benefits observed with SpO2 and PtcO2 are real improvements and were not due to a negative increase in RR or PtcCO2. There are also other associated benefits, such as the reduction in the length of stay in the NICU and the days of oxygen supply [57]. This technique is defined as safe since no worsening of intraventricular haemorrhages or development of periventricular leukomalacia has been observed. In addition, it was shown that this application does not cause stress or pain [52]. In this sense, the results of the study carried out on premature infants to find out the effect of Vojta therapy on bone modelling conclude that no participant showed signs of stress and pain during the intervention measured with NIPS [68]. In addition to these benefits related to respiratory aspects, it should be noted that Vojta therapy can also favour and promote motor development [69], which could be a very interesting benefit for the population of premature infants. This population, in addition to the respiratory distress, may have other comorbidities that can affect their motor development.

It is worth noting the difficulty to find more recent studies using Vojta Therapy in neonatology with premature infants and respiratory distress, besides those included in this review. On one hand, respiratory physiotherapy in neonatology remains controversial and, on the other hand, studies have long focused on CPT.

The study that used the AFE [50] technique did not analyse aspects related to respiratory function, but rather the safety of the technique related to the increase in gastroesophageal reflux episodes depending on the time of application. It concludes that no increase in reflux is observed at the proposed times. In this line, the study proposed by Bassani, et al. [70] in 2016, also defines it as a safe technique, since it did not affect cerebral blood flow in clinically stable premature infants.

Finally, PC was used in two studies [54,59], and the same description of the technique is observed in both. Each set consisted of three to four sustained chest compressions lasting approximately 5 s, followed by a slow, gentle ‘release phase’, with the chest wall fully released. The proposed improvements related to respiratory function (increase in SpO_2_, PaO_2_ and pulmonary re-expansion) are observed in the different groups proposed, with different interventions, so these improvements cannot be directly associated with this technique. In other studies, such as the one by Wong, et al. [71] in 2003, they do describe that the PC technique is more effective than CPT in lung re-expansion.

Regarding the intensity of the treatment, the duration of the intervention in the analysed studies has been around 10–20 min. Some authors [52,56] have not specified the specific time, and another [51] has stimulated the experimental group for 30 min (due to the combination of two techniques). It is observed that the average application time oscillates around 10 min [50,53,55,57]. Regarding the frequency, the number of daily sessions has also varied between one and three times. The same occurs with the intervention period; in the cases that define it, it is between 1 day and 2 weeks. Likewise, comparative studies are needed to determine suitability.

After analysing the methodological quality of the RCTs [50,51,53,54,57], the differences observed in the results obtained between the MINCIR scale and the PEDro scale are striking. Given the specificity of the PEDro scale for this type of study, it would be convenient to consider the value of the results obtained using this scale in the analysis of the methodological quality of the RCTs. Along these lines, three of the five RCTs [50,53,54] obtained a good methodological quality with some concern regarding the risk of bias, according to the ROB-2 scale. These concerns are related to the randomization process [50,53] and in the selection of the reported result [50,53,54]. On the other hand, El-Sharrawy, et al. [57] obtained a good methodological quality but a high risk of bias, due to the randomization process. It should be noted that the randomization of the study is not described as inadequate, rather it does not describe how the allocation was performed. Missing information is responsible for resulting in a score that indicates a high risk of bias, rather than lack of randomization. In addition, if the PEDro scale is analysed, all the items are scored as favourable except for items 2 and 3, related to assignment, and item 6, related to blinding of the therapist. Although information on randomization is missing, both the type of study and the results obtained are similar to other studies with adequate randomization [53,54], so we could extrapolate the results obtained. Likewise, item 6 (blinding of the therapist) is not observed in any of the studies. This is directly related to the study design, because physical therapists cannot be blinded to the techniques that they carry out hands-on. Therefore, it is an is an item that, despite not being achieved, cannot be improved. Finally, Gharu, et al. [51] presents a regular methodological quality, related to the assignment and blinding of the evaluator and a high risk of bias, mainly related to the lack of data on the results.

The MINCIR scale scores inadequate quality for all the articles with the exception of Oliveira et al. [56], who have similar scores to the rest of the studies in most items, except for item 2. It relates to the population studied, and it obtains a higher value given that the sample number is greater and also performs a calculation of the previous sample size. Another striking aspect of this scale is that all the articles obtain a similar score even though the type of study is different. This is due to the fact that the first domain, which analyses the design of the study, includes in the same score the clinical trial with simple blinding, without blinding, simple randomization and experimental studies (before and after).

Finally, based on the analysed results, CPT together with the stimulation of the RRo according to Vojta could be considered as a recommended technique to improve SpO_2_, RR and reduction of hospitalization time in the NICU and duration of hospitalization as well as oxygen delivery in preterm infants with RDS. On the other hand, both CPT, stimulation of the Vojta RRo and PC improve SpO_2_ and reduce atelectasis. In addition, it seems that the EFA technique is safe for premature infants with RDS.

A strength of this systematic review is the exhaustive search strategy as well as the use of the PRISMA 2020 [72] protocol in order to facilitate reproducibility and achieve the most possible rigorous and explicit scientific design. Another aspect to be positively assessed is the use of scales to measure methodological quality [15,16] and risk of bias [17,18], since no systematic reviews have been found on this subject with the use of these scales.

### 4.2. Limitations

Among the main limitations is the duration of some of the interventions (since in some of the studies, only one application was carried out), the follow-up of the results and the small sample size of some of the studies. The nature of the study itself limited control over treatment protocols and data collection or assessment. This means that many of the outcome measures were not homogeneous between the studies, which makes it difficult to draw conclusions from the review itself due to the limited comparability between the different data.

Another limitation could be the loss of articles due to the restriction of the language to English or Spanish and not having the full text. The articles lost in full text were published between 1983 and 1995, where there was no digitization of the texts as at present, an aspect that may have influenced the difficulty in accessing them.

### 4.3. Implications for Clinical Practice

In general, neonatal respiratory physiotherapy is related to bronchial cleansing; this is aimed at the elimination of secretions when they are already established. CPT is directly related to this aspect of respiratory physiotherapy. On the contrary, Vojta therapy aims to change the patient’s respiratory dynamics, creating an expansion of the rib cage, by improving lung diameters. As more air enters the lungs, the passive air outlet is also guaranteed, being able to prevent the accumulation of secretions. In addition, Vojta Therapy activates patterns of posture and movement that seem to be involved in improving the breathing pattern.

On the other hand, as can be deduced from the discussion, we considered the approach of applying techniques that are not only aimed at bronchial clearance interesting in premature infants with respiratory difficulties. Thus, the implementation of Vojta therapy is considered necessary, mainly through the stimulation of the chest zone according to Vojta, not only due to the results obtained in this review but also due to the analysis of the structural and functional difficulties of this population.

### 4.4. Implications for Future Research

We consider that there is still a lack of scientific evidence on the quality of intervention protocols in respiratory physiotherapy in NICUs with premature infants with RDS and carriers of some type of ventilatory support. This evidence would allow us to draw conclusions with a higher level of reliability. We would advise for future research to carry out RCTs with a larger number of premature babies, delimiting a specific range within prematurity (since respiratory difficulties are directly related to gestational age), with clearly differentiated controls, longer interventions and with greater monitoring of the results over time. This would allow to find out if these effects are maintained in the medium and long term. In addition, more studies are needed to specify the convenience of being able to combine different techniques according to the specific characteristics of the participants, as well as specify intensities, periods of application and benefits.

## 5. Conclusions

Positive effects were found with the intervention of the different respiratory physiotherapy techniques in relation to SpO_2_, RR, duration of oxygen therapy, compliance of the respiratory system, radiological findings of the chest and days of hospitalization. In addition, positive effects were also found in other parameters, measured in only one of the studies, such as HR, SA, partial pressure of oxygen, days of intervention, as well as the reduction in inspiratory and expiratory resistance.

CPT, RRo of Vojta, EFI and PC were the techniques found. These appear to be safe and may provide general benefits on impaired respiratory function in RDS. Despite not being able to determine a significant difference in the benefit of one of them over others, benefits in intragroup measures with their use stand out.

Although there are general benefits in respiratory function with the application of all the techniques, in the studies that make comparisons between groups, better results are observed regarding the days of hospitalization in the NICU, days of application of oxygen support [57] and RR [53] and SpO_2_ [51,53].

Based on the analyses carried out, it is likely that the most appropriate technique, among those analysed, is the application of Vojta reflex stimulation to intervene in the respiratory function of premature infants with respiratory difficulties such as respiratory distress syndrome, having taken into account the results obtained and the evidence of the studies.

The methodological quality of this review is mostly good according to the PEDro scale and inadequate according to the MINCIR scale. Regarding the risk of bias, mainly some concerns are observed in the analysed studies.

## Figures and Tables

**Figure 1 healthcare-11-01091-f001:**
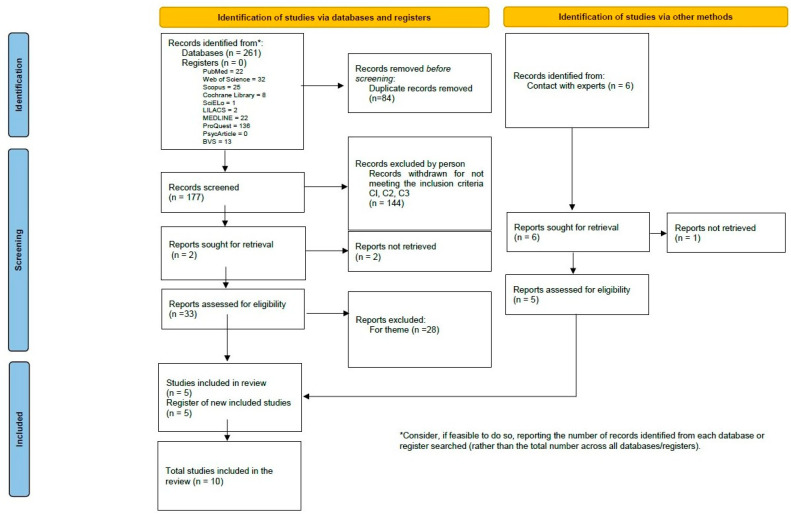
Flow diagram of the search results and screening for the identification of studies.

**Table 1 healthcare-11-01091-t001:** Characteristics of the physiotherapy treatments.

Study	Therapy	Age S-I (Days)	Period of Application	Frequency (Session/Day)	Intensity (Minutes)	Professional	Place
Camy, et al., 2011 [50]	EFI	G1 = 50.56 (SD = 7.92) G2 = 54.11 (SD = 23.02)	1	2	10	Physical therapist	NCU
Gharu, et al., 2016 [51]	Vojta (RRo + RC); CPT	NS	6	3	20 CG 30 GE	Physical therapist	NICU
Giannantonio, et al., 2010 [52]	RRo Vojta	G1 = 6 (SD = 1); G2 = 10.6 (SD = 3.5)	1	3	NS *	Physical therapist	NICU
Kaundal, et al., 2016 [53]	RRo Vojta; CPT	NS	14	3	10	Physical therapist	NCU
Kole, et al., 2014 [54]	RRo Vojta; CPT; PC	CG= 3.50 (SD = 1.36); EG1 = 3.55 (SD = 1.28); EG2 = 3.95 (SD = 1.15) y 3.95 (SD = 1.15)	14	3	20	Physical therapist	NICU
Metha, et al., 2016 [55]	CPT	9.55 (SD = 5.86)	1	1	10	Physical therapist	NICU
Oliveira, et al., 2019 [56]	CPT	NS	NS **	NS ***	NS	Physical therapist	NICU
El-Shaarawy, et al., 2017 [57]	RRo Vojta; CPT	1	NS ****	2	20	Physical therapist	NICU
Singh, et al., 2012 [58]	CPT	15 (SD = 7))	2	2	10	Physical therapist	Hospital
Wong, et al., 2006 [59]	PC	[2,3,4,5,6,7,8,9,10,11,12,13,14,15,16,17,18,19,20,21,22,23,24,25,26,27,28]	1	1	10	Physical therapist	NICU

S-I: Start intervention. EFI = Expiratory flow increase. CPT = Conventional chest physiotherapy. RRo = reflex rolling. RC = reflex creeping. CP = pulmonary compression. SD = Standard deviation. NICU = Neonatal intensive care unit. NCU = Neonatal care unit. CG = Control group. EG = Experimental group. NS: Not specified. *According to technique, around 4 min of stimulation, the article does not describe it. ** hospitalization time *** the physical therapist spends 6–8 h in the NICU during the week and decides on intervention according to characteristics **** Until discharge from the NICU.

**Table 2 healthcare-11-01091-t002:** Outcome measures and results.

Study	Outcome Measures	Statistical Significance (*p*)
Camy, et al., 2011 [50]	Reflux index	*p* = 0.245
Gharu, et al., 2016 [51]	SpO_2_	*p* < 0.001 IG
Giannantonio, et al., 2010 [52]	RR	NS
	SpO_2_	G1 < 0.001 IG; G2 < 0.05 IG
	PtcCO_2_	NS
	PtcO_2_	G1 < 0.0001 IG; G2 < 0.01 IG
	NIPS	No pain or stress
	PIPP	No stress
	Cerebral ultrasound scans	No worsening
Kaundal, et al., 2016 [53]	SpO_2_	*p* ≤ 0.05 IG
	RR	*p* ≤ 0.05 IG
Kole, et al., 2014 [54]	SpO_2_	*p* < 0.001 IG
	PaO_2_	*p* < 0.001 IG
	Duration of oxygen therapy	*p* = 0.012 EG
	Chest X-ray	Descriptive data
	Number of days of intervention	*p* = 0.041 EG
Mehta, et al., 2016 [55]	HR	*p* = 0.01 UG
	RR	*p* < 0.0001 UG
	SA	*p* < 0.0001 IG
	SpO_2_	*p* < 0.0001 UG
	Auscultation	*p* < 0.01 UG
	Chest X-ray	*p* < 0.0001 UG
Oliveira, et al., 2019 [56]	Hospitalization in the NICU	*p* = 0.39 BG
	Hospitalization out of the NICU	*p* = 0.77 BG
	Duration of invasive mechanical ventilation	*p* = 0.04 BG
	Duration of non-invasive mechanical ventilation	*p* < 0.01 BG
	Duration of oxygen therapy	*p* = 0.31 BG
	Duration of CPAP	*p* = 0.04 BG
	Duration of non-invasive mechanical ventilation + duration of CPAP	*p* < 0.01 BG
El-Shaarawy, et al., 2017 [57]	SpO_2_ RR Hospitalization in the NICU Duration of oxygen therapy	*p* = 0.377 BG
		*p* = 0.001 IG
		*p* = 0.863 BG
		*p* = 0.001 IG
		*p* = 0.021 BG
		*p* = 0.005 BG
Singh, et al., 2012 [58]	Respiratory system compliance	*p* < 0.0001 IG
	Decreased inspiratory resistance	*p* < 0.0001 IG
	Reduced expiratory resistance	*p* < 0.0001 IG
	Reinflation of collapsed lung	*p* < 0.0001 IG
Wong, et al., 2006 [59]	Respiratory system compliance	*p* = 0.023 IG
	Respiratory system resistance	*p* = 0.147 IG

SpO_2_ = Oxygen saturation; RR = Respiratory rate; PtcCO_2_ = Transcutaneous carbon dioxide pressure; PtcO_2_ = Transcutaneous oxygen pressure; NIPS = Neonatal Infant Paint Scale; PIPP = Premature Infant Paint Profile; PaO_2_ = Partial pressure of oxygen; HR = Heart rate; SA = Silverman Anderson score; CPAP = Continuous Positive Airway Pressure; BG = Between groups; IG = Intra groups; NS = Not specified, the change is described as not significant but does not give you the *p* value; UG = Union of groups, the results are regarding the change when applying the treatment in all users compared to the baseline; G1 = Group 1; G2 = Group 2.

**Table 3 healthcare-11-01091-t003:** Methodological quality assessed with the MINCIR Therapy scale.

MINCIR Therapy Scale	D1	D2	D3.1	D3.2	D3.3	D3.4	TS	MQ
Camy, et al., 2011 [50]	6	1	2	2	3	1	15	INAD
Gharu, et al., 2016 [51]	6	2	2	2	3	1	16	INAD
Giannantonio, et al., 2010 [52]	6	2	2	1	3	1	15	INAD
Kaundal, et al., 2016 [53]	6	2	2	2	3	1	16	INAD
Kole, et al., 2014 [54]	6	2	2	2	3	1	16	INAD
Mehta, et al., 2016 [55]	6	2	2	2	3	1	16	INAD
Oliveira, et al., 2019 [56]	3	10	2	3	2	3	23	ADE
El-Shaarawy, et al., 2017 [57]	6	2	2	2	3	1	16	INAD
Singh, et al., 2012 [58]	6	2	2	2	2	1	15	INAD
Wong, et al., 2006 [59]	6	2	2	1	2	3	16	INAD

D1 = Research design; D2 = Studied population “x” justification factor (×2); D3.1= Methodology. Objective; D3.2 = Methodology. Design; D3.3 = Methodology. Sample selection criteria; D3.4 = Methodology. Sample size; TS = total score; MQ = Methodological quality; INAD = inadequate; ADE = adequate.

**Table 4 healthcare-11-01091-t004:** Methodological quality assessed with the PEDro scale.

PEDro Scale	1	2	3	4	5	6	7	8	9	10	11	T
Camy, et al., 2011 [50]		X			X			X	X	X	X	6
Gharu, et al., 2016 [51]		X			X			X	X	X		5
Kaundal, et al., 2016 [53]		X			X		X	X	X	X	X	7
Kole, et al., 2014 [54]		X	X		X		X			X	X	6
El-Shaarawy, et al., 2017 [57]	X			X	X		X	X	X	X	X	7

1 = Eligibility criteria were specified; 2 = Random allocation; 3 = Allocation was concealed; 4 = The groups were similar at baseline; 5 = There was blinding of all subjects; 6 = There was blinding of all therapists; 7 = There was blinding of all assessors; 8 = Measures of at least one key outcome were obtained from more than 85% of the subjects initially allocated to groups; 9 = All subjects for whom outcome measures were available received the treatment or control condition as allocated or where this was not the case, data for at least one key outcome were analysed by “intention to treat”; 10 = The results of between-group statistical comparisons are reported for at least one key outcome; 11 = The study provides both point measures and measures of variability for at least one key outcome; T = Total score; X = Meets the criteria.

**Table 5 healthcare-11-01091-t005:** Methodological quality assessed with the NOS scale.

NOS Scale	Selection				Comparability	Exposure			Total Stars	Conclusion
	1. Is the case definition adequate? (a *, b, c)	2. Representativeness of the cases (a *, b)	3. Selection of controls (a *, b, c)	4. Definition of controls (a *, b)	5. Comparability cases-controls (a *, b *)	6. Verification of exposition (a *, b *, c, d)	7. Verification cases and controls (a *, b)	8. Non-response rate (a *, b, c)		
Oliveira, et al., 2019 [56]	a *	a *	b	a *	a * y b *	d	a *	a *	7	Low risk of bias

1. (a) yes, with independent validation *; (b) yes, e.g., record linkage or based on self-reports; (c) no description. 2. (a) consecutive or obviously representative series * (b) potential for selection bias or unreported. 3. (a) community controls *; (b) hospital controls; (c) no description. 4. (a) no history of disease *; (b) no source description. 5. (a) study the controls for __ (Select the most important factor) *; (b) study controls for any additional factors *. 6. (a) secure registration *; (b) structured interview where case/control status is not seen *; (c) interview not blinded to case/control status; (d) written self-report or medical record only; (e) no description. 7. (a) yes *; (b) not. 8. (a) the same rate for both groups *; (b) those who did not respond described; (c) different rate and without designation.

**Table 6 healthcare-11-01091-t006:** Risk of bias assessed with the ROB-2 Cochrane scale.

Study	Randomization Process	Deviations from Intended Interventions	Missing Outcome Data	Measurementof the Outcome	Selection of the Reported Result	Overall
Camy, et al., 2011 [50]	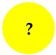	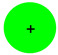	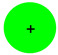	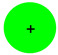	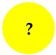	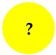
Gharu, et al., 2016 [51]	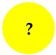	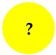	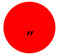	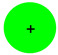	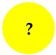	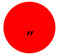
Kaundal, et al., 2016 [53]	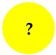	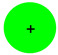	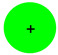	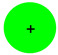	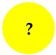	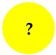
Kole, et al., 2014 [54]	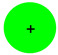	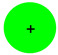	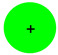	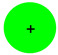	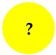	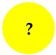
El-Shaarawy, et al., 2017 [57]	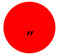	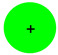	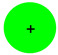	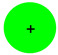	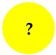	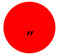

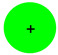
 Low risk. 
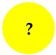
 Some concerns. 
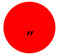
 High risk.

## Data Availability

Data supporting the findings of this study are available from the lead author, (IB-A), upon reasonable request.

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
