# Peer review of "Effects of Chest Physiotherapy in Preterm Infants with Respiratory Distress Syndrome: A Systematic Review"

_healthcare, 2023, doi:10.3390/healthcare11081091_

Round 1

Reviewer 1 Report

Thank you for allowing me to review your manuscript entitled “Effects of Chest Physiotherapy in Preterm Infants With Respiratory Distress Syndrome: A Systematic Review”.

The manuscript focuses on a subject of vital significance for the quality of life in premature children such as the treatment of respiratory problems using respiratory physiotherapy. taking into account the variability of the techniques used, the authorss have carried out a structured review according to the prisma standards to analyze the evidence and try to establish some conclusions on the recommendations for respiratory physiotherapy techniques in premature children.

However, some modifications and clarifications are necessary to accept its publication:

1) Please, adjust the number of words and structure of the Abstract to the norms indicated in the journal:

  • Abstract: The abstract should be a total of about 200 words maximum. The abstract should be a single paragraph and should follow the style of structured abstracts, but without headings: 1) Background: Place the question addressed in a broad context and highlight the purpose of the study; 2) Methods: Describe briefly the main methods or treatments applied. Include any relevant preregistration numbers, and species and strains of any animals used. 3) Results: Summarize the article's main findings; and 4) Conclusion: Indicate the main conclusions or interpretations. The abstract should be an objective representation of the article: it must not contain results which are not presented and substantiated in the main text and should not exaggerate the main conclusions.

2) Review keywords; some do not appear in the MeSH.

3) Adjust the bibliographical references in the text to the journal's standards:

In the text, reference numbers should be placed in square brackets [ ], and placed before the punctuation; for example [1], [1–3] or [1,3]. For embedded citations in the text with pagination, use both parentheses and brackets to indicate the reference number and page numbers; for example [5] (p. 10). or [6] (pp. 101–105).

4) Page 3, 2.5, line 109. Specify the meaning of the initials MINCIR before them..

5) Page 4, line 154. Specify the number of participants in the included studies except for the Metha et al. Why? Did the authors not describe this data? Please add or specify that it has not been provided for this study.

6) Page 4, line 153-155. Review this sentence, it is not clear what it means. Does it mean that in some studies the control groups received the same intervention as the experimental group and in other studies they received another intervention?

7) Page 5, line 167. Shows the end of a parenthesis, but not the start. Clarify this section

8) Page 5, line 167-168. Appointment Conventional chest physiotherapy What does it consist of? What techniques does it include? Subsequently, you cite them in the Discussion section, but it is convenient that he also describe them in section 3.4 of the Results section.

9) Page 5, Table 1. Why are the session frequencies and intensity of the Giannantonio et al. study highlighted? What does SD mean in age? Standard deviation?

10) Page 6, Table 2. In the outcome measures column, indicate only the acronyms of the variables and describe their meaning below the table. For example, instead of Heart rate (HR) just specify HR.

11) In point 3.7, line 279 mentions CPAP, as in table 2, but does not describe its meaning. Please, indicate it under table 2 along with the meaning of the other acronyms.

12) In the Discussion section there is a paragraph (lines 361-64) that should go under Conclusions.

13) In section 4.4 there is a spelling error, line 462. “evidencewould”. Please correct it.

14) Check references according to journal guidelines:
References should be described as follows, depending on the type of work:

 Journal Articles:
1. Author 1, A.B.; Author 2, C.D. Title of the article. Abbreviated Journal Name YearVolume, page range.

Books and Book Chapters:
2. Author 1, A.; Author 2, B. Book Title, 3rd ed.; Publisher: Publisher Location, Country, Year; pp. 154–196.
3. Author 1, A.; Author 2, B. Title of the chapter. In Book Title, 2nd ed.; Editor 1, A., Editor 2, B., Eds.; Publisher: Publisher Location, Country, Year; Volume 3, pp. 154–196.

Author Response

Mr. Reviewer,
Thank you very much for your time and input in the review. Below I attach a descriptive word responding to all your contributions. I also attach the manuscript with all reviewers' modifications.

I remain at your disposal for any further clarification.

Ana Igual Blasco

Response to Reviewer 1 Comments

Point 1: Please, adjust the number of words and structure of the Abstract to the norms indicated in the journal

Response 1: The number of words has been reduced from 217 to 199. The abstract is considered to contain the proposed structure, in a single paragraph, structured without a heading. I understand that when considering without headers I should not add the words antecedents, methods, etc. I have reviewed publications in Healthcare magazine and have seen summaries with these statements and others without them. I ask that if you need any new changes, let me know again.

Abstract: Preterm birth carries a higher risk of respiratory problems. The objective of the study is to summarize the evidence on the effect of chest physiotherapy in the treatment of respiratory difficulties in preterm infants and to determine which is the most appropriate technique, and whether they are safe. Searches were made in PubMed, WOS, Scopus, Cochrane Library, SciELO, LILACS, MEDLINE, ProQuest, PsycArticle and VHL until 04/30/2022. Eligibility criteria were: study type, full text, language, and treatment type. No publication date restrictions were applied. The MINCIR Therapy and PEDro scales were used to measure methodological quality and the Cochane risk of bias and Newcastle Ottawa quality assessment Scale to measure risk of bias. We analysed 10 studies with 522 participants. The most common interventions were conventional chest physiotherapy and stimulation of the pectoral point of Vojta. Lung compression and increased expiratory flow was also used. Heterogeneities were observed regarding the duration of the interventions and the number of participants. The methodological quality of some articles was not adequate. All techniques were shown to be safe. Benefits were described after conventional chest physiotherapy, Vojta reflex rolling, and lung compressions interventions. Improvements by Vojta's reflex rolling are highlighted in the comparative studies.

Point 2: Review keywords; some do not appear in the MeSH.

Response 2: The keywords in the MeSH have been revised and the keyword "respiratory distress" has been removed and chest physiotherapy has been modified by Physical therapy modalities. The following keywords have been left: “Respiratory distress syndrome; Physical therapy modalities; premature and neonatal intensive care unit.”

Point 3: Adjust the bibliographical references in the text to the journal's standards:

In the text, reference numbers should be placed in square brackets [ ], and placed before the punctuation; for example [1], [1–3] or [1,3]. For embedded citations in the text with pagination, use both parentheses and brackets to indicate the reference number and page numbers; for example [5] (p. 10). or [6] (pp. 101–105).

 Response 3: Superscripts have been changed to square brackets. See modification in manuscript.

Point 4: Page 3, 2.5, line 109. Specify the meaning of the initials MINCIR before them.

Response 4: The acronym MINCIR comes from the Spanish group that created the scale "Metodología e Investigación en Cirugía", it has been modified in the text, leaving: “The Methodology and Research in Surgery (MINCIR) scale”.

Point 5: Page 4, line 154. Specify the number of participants in the included studies except for the Metha et al. Why? Did the authors not describe this data? Please add or specify that it has not been provided for this study.

Response 5: The text does mention the number of participants from Metha, et al. This data can be found at the beginning of point 3.3 on line 141-143. “Metha, et al. [55] had two groups with the same intervention where the difference between the groups was a medical situation, each group had 30 participants.”

Point 6: Page 4, line 153-155. Review this sentence, it is not clear what it means. Does it mean that in some studies the control groups received the same intervention as the experimental group and in other studies they received another intervention?

Response 6: This paragraph refers to the fact that some studies have used an intervention modality (in this case CPT) as a control, and in other studies, this same intervention modality (CPT) has been used as an experimental intervention modality. The paragraph has been modified for better understanding: “It should be noted that the control group of the last 4 studies mentioned received an intervention modality that in the studies by Metha, et al. [55], Oliveira, et al. [56] and Singh, et al. [58] was considered as an experimental intervention modality”

Point 7: Page 5, line 167. Shows the end of a parenthesis, but not the start. Clarify this section

Response 7: A comma has been changed for opening parentheses. “one [54] has two experimental groups (one group with Vojta's reflex turning and the other lung compression).”

Point 8: Page 5, line 167-168. Appointment Conventional chest physiotherapy What does it consist of? What techniques does it include? Subsequently, you cite them in the Discussion section, but it is convenient that he also describe them in section 3.4 of the Results section.

Response 8:  Added this clarification “In these studies that used CPT, the techniques were not homogeneous, although they all applied vibration and percussion techniques.”

Point 9: Page 5, Table 1. Why are the session frequencies and intensity of the Giannantonio et al. study highlighted? What does SD mean in age? Standard deviation?

Response 9: Removed underlining from the frequency and intensity of the study by Giannantonio, et al. because it was a mistake. SD does mean standard deviation. Added to the table legend.

Point 10: Page 6, Table 2. In the outcome measures column, indicate only the acronyms of the variables and describe their meaning below the table. For example, instead of Heart rate (HR) just specify HR.

Response 10: The table has been modified based on this recommendation. To see in manuscript.

Point 11: In point 3.7, line 279 mentions CPAP, as in table 2, but does not describe its meaning. Please, indicate it under table 2 along with the meaning of the other acronyms.

Response 11: Continuous Positive Airway Pressure (CPAP). Added to table 2.

Point 12: In the Discussion section there is a paragraph (lines 361-64) that should go under Conclusions.

Response 12: This paragraph has been moved to conclusions, see in manuscript.

Point 13: In section 4.4 there is a spelling error, line 462. “evidencewould”. Please correct it.

Response 13: spelling error has been changed: “evidence would”

Point 14: Check references according to journal guidelines:
References should be described as follows, depending on the type of work:

 Journal Articles:
1. Author 1, A.B.; Author 2, C.D. Title of the article. Abbreviated Journal Name Year, Volume, page range.

Books and Book Chapters:
2. Author 1, A.; Author 2, B. Book Title, 3rd ed.; Publisher: Publisher Location, Country, Year; pp. 154–196.
3. Author 1, A.; Author 2, B. Title of the chapter. In Book Title, 2nd ed.; Editor 1, A., Editor 2, B., Eds.; Publisher: Publisher Location, Country, Year; Volume 3, pp. 154–196.

Response 14: References have been reviewed and modified based on recommendations. see in manuscript.

Reviewer 2 Report

The present manuscript is a systematic review study aiming to identify the most appropriate technique of chest physiotherapy in the treatment of respiratory difficulties in preterm infants.

After the systematic study of data, MINCIR Therapy and PEDro scales were apply to assess methodological quality, and the Cochane risk of bias and Newcastle Ottawa quality assessment Scale to measure risk of bias.

The review analyzed 10 studies with 522 participants.

The authors make an accurate analysis of each reports and their interactive consequences, with clear descriptions of the analysis in relation to the most relevant criteria to decipher the clinical efficiency of physiotherapy interventions.

Clear arguments support the discussion and conclusion.

They conclude with a clear message, whit potential therapeutic consequences:

“the most appropriate technique, among those analyzed, is the application of Vojta reflex stimulation to intervene in the respiratory function of premature infants with respiratory difficulties such as respiratory distress syndrome, having taken into account the results obtained and the evidence of the analyzed studies”.

Limitations of the study and implications in clinical practice and research have been clearly described and discussed with arguments in specific sections.

Regarding the scientific interest of reviewed topic, the physical therapy in Neonatal Respiratory Distress Syndrome (RDS) is a condition of perinatal pulmonary insufficiency. Therapeutic measures are necessary because the child may even die from progressive hypoxia and respiratory failure. RDS is due to a deficit of alveolar surfactant together with a structural immaturity of the lung. In preterm birth, babies present structural and functional immaturity. The incidence is around 60% of those babies under 28 weeks of gestation (wg) and less than 5% of those over 34 wg. The goal of RDS treatment is to provide interventions to maximize survival while minimizing potential adverse effects, including Bronchopulmonary Dysplasia (BPD).

Flow diagram and tables are descriptive and representative of analyzed data. 

Author Response

Mr. Reviewer,

Thank you very much for your time and input in the review.

Ana Igual Blasco

Reviewer 3 Report

The study approaches a critical situation of the newborn due to the lack of surfactant and lung immaturity, mainly the reduced thinning of the capillary-alveolar barrier. The data are also important because it compares different techniques of chest physiotherapy.

Author Response

(The authors gave the same response as above.)

Reviewer 4 Report

This research targets a difficult and broad topic. Respiratory physiotherapy in premature babies within NICU environment is very controversial and a description of the current state of evidence is very relevant. This research analysed 2 different approaches according to their goals: airway clearance and ventilatory pattern. This point of view is interesting, since voluntary respiratory exercises or training is not available in this cohort as it is in older children or adults. Also, postural control seems to be the base for an efficient respiratory capacity, but this is inmature in premature and underweight babies, in comparison with those born at adequate gestational age.

Authors assessed and broke down very clearly the methodological quality, as well as the risk of bias of the available literature. Their conclusion is correctly based on their findings, and states the need of further quality research and therapy development in this field (which is poor as described in this work). This is probably limited also due to its specificity.

Reflex Locomotion or Vojta Therapy is an arising topic in the physiotherapy / rehabilitation literature regarding premature children with and without neurological conditions. Restrictive respiratory conditions are often observed in this cohort, and active techniques aiming the improvement of the ventilatory pattern, as well as prevention of further respiratory conditions, require investigation.

Although the transfer of these conclusions to the clinical field is still limited and requires further discussion, this work summaries the state of evidence of the wide/diverse chest physiotherapy approaches within the scientific literature. On top of setting the current baseline, this work points towards the development of specific interventional protocols in neonatology care, for both clinic and research field.

Please read my comments to this article on the commented PDF file attached

Author Response

Thank you very much for your time and input in the review. Below I attach the PDF that he sent me with his contributions, responding to each comment. I also attach the manuscript with all reviewers' modifications.

I remain at your disposal for any further clarification.

Ana Igual Blasco
